# Postoperative Radiological Improvement after Staged Surgery Using Lateral Lumbar Interbody Fusion for Preoperative Coronal Malalignment in Patients with Adult Spinal Deformity

**DOI:** 10.3390/jcm12062389

**Published:** 2023-03-20

**Authors:** Akihiko Hiyama, Daisuke Sakai, Hiroyuki Katoh, Masato Sato, Masahiko Watanabe

**Affiliations:** Department Orthopaedic Surgery, Tokai University School of Medicine, 143 Shimokasuya, Isehara 259-1193, Japan

**Keywords:** adult spinal deformity, lateral lumbar interbody fusion, coronal malalignment, coronal balance distance, spinopelvic parameters

## Abstract

This retrospective observational study evaluated improvement in coronal malalignment (CM) after anteroposterior staged surgery using lateral lumbar interbody fusion (LLIF) in patients with coronal lumbar curve adult spinal deformity (ASD). Sixty patients with ASD underwent surgery; 34 had SRS–Schwab type L lumbar curve. Patients with a coronal balance distance (CBD) ≥20 mm were diagnosed with CM. Using the Obeid CM classification, we classified the preoperative coronal pattern as concave CM (type 1) or convex CM (type 2). Demographic, surgical, and radiological parameters were compared. Whole-spine standing radiographs were assessed preoperatively and postoperatively. Twenty-three patients had type 1A, six had type 2A, five had no CM, and none had type 1B or 2B according to the Obeid CM classification. Compared with patients with Obeid type 1A, those with Obeid type 2A had significantly higher preoperative and postoperative coronal L4 tilts and a smaller change in corrected CBD (Δ|CBD|) (76.6 mm vs. 24.1 mm, *p* < 0.001). At the final follow-up, 58.6% (17/29 patients) of patients with SRS–Schwab type L CM showed improvement after corrective fusion using LLIF. Although the difference was not statistically significant, CM improved in 69.6% (16/23 patients) of patients with Obeid type 1A type but only 16.7% (1/6 patients) of those with Obeid type 2A type (*p* = 0.056). CM was more likely to remain after anteroposterior staged surgery using LLIF in patients with preoperative Obeid type 2A ASD.

## 1. Introduction

Adult spinal deformity (ASD) is one of the most disabling diseases, and the incidence of this condition is 32% in adults and increases to 60% in the elderly population [1].

The quality of life (QOL) is lower in people with ASD than in those with other debilitating chronic diseases [2]. Surgery may be viable for people with ASD when nonsurgical management fails. Corrective surgery for ASD is associated with a significantly improved QOL compared with nonsurgical management [3,4]. The goal of surgical intervention in patients with ASD is to improve pain and QOL by correcting the balance of the sagittal and coronal spine.

In the past decade, surgical efforts have been directed to correct sagittal spinopelvic parameters, and coronal malalignment (CM) has received less attention. However, recent studies suggest that CM causes pain and disability and has functional outcomes [5,6,7]. Given this background, Bao et al. evaluated the classification of coronal imbalance and divided patients into groups based on whether their C7 plumb line (C7 PL) was neutral (type A), shifted toward the concavity (type B), or shifted toward the convexity (type C) of their scoliosis [8]. These authors reported that patients with type C curves were more likely to have postoperative CM and poor clinical outcomes. Obeid et al. advanced this classification system to help provide surgical recommendations for deformity correction in adult patients with scoliosis. They focused on the CM scheme described by Bao and colleagues and added modifiers related to the relative flexibility of the primary and fractional lumbosacral curves [9]. Despite these assessment methods, treating CM remains a challenging surgical problem because this deformity is often associated with the sagittal imbalance that requires tridimensional corrective maneuvers. Over the years, many techniques have been suggested as surgical solutions, but the topic remains controversial.

The lateral lumbar interbody fusion (LLIF) procedure is a minimally invasive surgery that offers several potential advantages over traditional posterior approaches, including reduced blood loss and less postoperative pain [10,11,12,13]. In recent years, the usefulness of corrective surgery, such as the LLIF procedure, has been applied in patients with ASD. However, many things still need to be researched about its usefulness in ASD patients with CM [14,15,16]. Therefore, this study aimed to evaluate the utility of the LLIF procedure for correcting CM deformity in patients with coronal lumbar curve ASD.

## 2. Materials and Methods

### 2.1. Ethics Approval

The ethical committee of our hospital approved this study (22R-217), and the guidelines and regulations of the ethics review board were followed for all methods. As this study was retrospective, the requirement for informed consent was waived.

### 2.2. Included Patients

We retrospectively collected the data for consecutive patients aged > 40 years who had undergone ASD correction between January 2016 and November 2021. The main indication for ASD surgery is severe pain associated with curve progression making it difficult to stand for long periods. Additionally, the pain must be resistant to conservative treatment, such as medications, braces, and exercises. Three board-certified spinal surgeons performed the procedures. Patients were considered candidates for deformity correction long spinal fusion, indicated for ASD, if an entire course of conservative care had been exhausted. For adult patients aged >40 years, the radiographic inclusion criteria were at least one of the following: Cobb angle ≥ 20°, the sagittal vertical axis (SVA) ≥ 5 cm, pelvic tilt (PT) ≥ 25°, and thoracic kyphosis (TK) ≥ 60°. The Scoliosis Research Society–Schwab (SRS–Schwab) adult deformity classification was recorded for each patient [17].

### 2.3. Bone Density Evaluation

We measured Hounsfield unit (HU) values obtained from the preoperative CT scans of the upper instrumented vertebra (UIV), UIV + 1, and UIV + 2 according to the methodology described in our previous paper [18]. Briefly, HU values were measured in three sections: immediately inferior to the superior end plate, in the middle of the vertebral body, and immediately superior to the inferior end plate in preoperative CT images. The preoperative axial CT image layer for HU measurement visualizes the end plate region’s most prominent cancellous and cortical bone areas. The region of interest (ROI) was chosen manually to fit the shape of each structure. After establishing a consistent and appropriate ROI, the Picture Archiving and Communication system calculated the mean HU values. Finally, we evaluated the average HU values calculated in the three sections.

### 2.4. Surgical Procedure

We have previously described the surgical procedures in detail [19,20,21]. The anteroposterior staged surgery used to correct ASD was performed as follows. Multilevel LLIF using the oblique lateral interbody fusion (Medtronic, Minneapolis, MN, USA) or extreme lateral interbody fusion (NuVasive Inc., San Diego, CA, USA) was performed from L1/2 or L2/3 to L4/5, and this was followed by posterior instrumentation. Anterior release by LLIF was performed at the surgeon’s discretion. Autologous iliac crest bone and artificial bone made of hybridized hydroxyapatite and type I collagen (ReFit^®^, HOYA Technosurgical Co., Tokyo, Japan) or demineralized bone matrix (Grafton^®^, Medtronic, Dublin, Ireland) were mixed and inserted into the PEEK cage. Since they became available in February 2020, we have been using titanium cages (NuVasive). One week later, posterior corrective fusion and transforaminal lumbar interbody fusion at L5/S1 were performed using a pedicle screw system. The posterior approach was performed for Schwab grade 1 or 2 osteotomies from L1/2 to L5/S1. L5/S1 transforaminal lumbar interbody fusion was then routinely performed using large lordotic cages, and lumbar lordosis (LL) was restored using a rod cantilever and compression technique. Except for one patient, the lower instrumented vertebra (LIV) was up to the iliac, and we commonly used a sacral alar–iliac screw with a diameter of 8.5 mm and a length of ≥80 mm.

### 2.5. Data Collection

The data comprised radiological and demographic information and surgical parameters. All patients underwent full-length anteroposterior and lateral standing X-rays. Radiographs were evaluated for coronal alignment and sagittal spinopelvic parameters. These measures were used to classify each patient according to the SRS–Schwab classification. Briefly, sagittal measurements included (1) LL; (2) TK (T5–T12); (3) PT; (4) pelvic incidence (PI); (5) sacral slope; (6) PI–LL; and (7) SVA. The following parameters were measured in the coronal plane. Coronal balance distance (CBD) was defined as the horizontal distance between the C7 PL and the central sacral vertical line. C7 PL shifted to the right and was defined as positive and to the left as negative. Δ|CBD| was calculated from postoperative CBD−preoperative CBD and calculated as its absolute value. That is, |CBD| is represented as a positive. For example, if the preoperative CBD was 70.1 mm and the postoperative CBD was −20.3 mm, −20.3 − 70.1 = −90.4 mm and Δ|CBD| = 90.4mm. The major Cobb angles were defined as the angle between the superior end plate of the most tilted vertebra cranially and the inferior end plate of the most tilted vertebra caudally. L4 coronal tilt and L5 tilt were defined as the angle between the superior end plate of L4 or L5 and the horizontal line [22]. We also evaluated each patient’s curve flexibility in the coronal plane using side bending and traction radiographs. The SRS–Schwab system includes an assessment of the coronal curve type and sagittal modifiers [17]. The coronal curve type is based on the spine location and Cobb angle (>30°) of the scoliotic curves and is classified as follows: (1) curve type T: thoracic only; the thoracic major curve of >30° at the apical level of T9 or higher; (2) curve type L: thoracolumbar/lumbar only; isolated thoracolumbar or lumbar curve > 30° at the apical level of T10 or lower; (3) curve type D: a double major curve with thoracic and thoracolumbar/lumbar curve of >30°; and (4) curve type N: normal; no definite coronal deformity (all coronal curves < 30°). After classifying by type of coronary curve, we included patients with type L curves in this study. All patients were subgrouped according to their preoperative coronal alignment as defined by Obeid et al. according to the distance between their C7 PL and their central sacral vertical line [9]. The Obeid CM classification system first divides the patients with CM into two main types according to their CM deformity patterns. Concave CM (type 1) is defined as CM with a coronal T1 PL falling at the side of the concavity of the main coronal curve. By contrast, convex CM (type 2) is defined as CM with a coronal T1 PL falling at the side of the convexity of the main coronal curve > 20 mm (Figure 1).

Patients can be subtyped further as type 1A, having the main curve apex between T12 and L4, and type 1B, having the apex above T12. Type 1A1 is flexible, and type 1A2 is rigid. Type 2A has the apex of the main curve between T12 and L4, whereas type 2B has the apex below L4. Type 2A1 has a normal lumbosacral junction, and type 2A2 has a degenerative lumbosacral junction. Patients with no CM are defined as having a C7 PL within 20 mm of the CBD. Pre- and the postoperative coronal imbalance were defined as |CBD| ≥ 20 mm [9]. We defined the proximal junctional angle (PJA) as the sagittal Cobb angle between the inferior endplate of UIV and the superior endplate of UIV + 2. For PJA, the value at the time of the last observation was used, but for patients with proximal junctional failure (PJF), the value at the time of PJF observation was used [20,21,23]. These data were then compared between the patients with Obeid type 1A and those with Obeid type 2A CM.

### 2.6. Statistical Analysis

The data were analyzed using IBM SPSS Statistics (version 23.0; IBM Corp., Armonk, NY, USA). All values are expressed as the mean ± standard deviation. We first used the Kolmogorov–Smirnov test for all continuous variables to test for normal distributions. Next, we investigated the relationships between two groups using the chi-squared test for categorical variables, *t* test analysis of variance for continuous variables, and the Mann–Whitney *U* test. The type 1 error was set at 5% for all statistical analyses, and *p* < 0.05 was assumed to be significant.

## 3. Results

A total of 60 ASD patients (2 men and 58 women) underwent surgery. According to the SRS–Schwab Adult Deformity Classification, 34 patients had the type L coronal curve type, 2 patients had the double coronal curve type, and 24 patients had the normal coronal curve type. Patients with SRS–Schwab type L CM were classified using the Obeid CM classification system (Table 1).

The distribution was type 1A in 23 patients, 2A in 6 patients, and 1B or 2B in no patients. In the subanalysis, 7 patients were classified as having type 1A1, 16 having type 1A2, 2 having type 2A1, and 4 having type 2A2 CM. CM was not classified in five patients. The overall findings were assessed according to demographic data, radiographic measurements, and surgical data for twenty-three Obeid type 1A and six Obeid type 2A patients and are presented in Table 2.

The mean age at the time of surgery was 71.8 (±6.5) years (range 55–84 years), and the average follow-up period was 26.1 (±9.9) months. Sex, age, height, body weight, body mass index at the time of surgery, CBD shift type, the apex of the main curve, instrumented levels, distribution of UIV or LIV, interbody fusion levels, HU values, operation time, estimated blood loss, and length of hospital stay did not differ significantly between the Obeid type 1A and Obeid type 2A groups. The number of patients with major complications were as follows: five patients (17.2%) with PJF and eight patients (27.6%) with rod breakage (RB). Eight patients (27.6%) required revision surgery because of mechanical failure. These complications were Obeid type 1A patients (Table 2).

In the examination of the radiographic sagittal parameters across the entire cohort, we found significant improvements in PI–LL, including SVA, LL, and PT. However, TK and PJA increased postoperatively. The major Cobb angle was corrected markedly from 48.8° preoperatively to 13.2° postoperatively (*p* < 0.001). L4 coronal tilt improved significantly from 18.6° preoperatively to 8.7° postoperatively (*p* < 0.001) and L5 coronal tilt from 12.9° preoperatively to 7.3° postoperatively (*p* < 0.001). The major Cobb angle and L4 and L5 coronal tilts also improved in both groups. However, the L4 and L5 coronal tilt were larger before and after surgery in the Obeid type 2 group than in the Obeid type 1A group. The │CBD│ did not differ significantly between the type 1A and type 2A groups before and after surgery. In patients with Obeid type 1A, the │CBD│ changed from 78.4 mm before to 22.8 mm after surgery (*p* < 0.001). In patients with Obeid type 2A, the change in the │CBD│ from 46.9 mm to 38.5 mm was not significant (*p* = 0.506) (Table 3).

Analysis of the change or correction (Δ) for each spinal parameter showed a significant difference in ΔTK for the sagittal parameter TK. ΔTK was larger in patients with Obeid type 2A, but other sagittal parameters did not differ significantly between groups. Δ│CBD│ was significantly larger in patients with Obeid type 1A than those with Obeid type 2A (76.6 mm vs. 24.1 mm; *p* < 0.001) (Table 4).

There were no significant intergroup differences in the changes in the major Cobb angle, correction of coronal tilts of L4 and L5, and the PJA (Table 4). CM was found in 29 of 34 patients with SRS–Schwab type L ASD preoperatively and remained in 12 of these patients at the final follow-up. At the final follow-up, 58.6% (17/29) of patients with SRS–Schwab type L CM showed improvement in CM after corrective fusion using LLIF. Although the difference was not statistically significant, CM improved in 69.6% of patients with Obeid type 1A CM but only 16.7% of those with Obeid type 2A CM (*p* = 0.056) (Table 5).

Typical postoperative changes are shown in Figure 2, Figure 3 and Figure 4.

## 4. Discussion

The current study retrospectively evaluated ASD patients who underwent corrective surgery with anteroposterior staged surgery using LLIF. A total of 29 of the 34 patients with SRS–Schwab type L ASD with lumbar scoliosis had a CM with CBD ≥ 20 mm. We also found that compared with patients with Obeid type 1, those with Obeid type 2 had significantly greater preoperative L4 and L5 tilts and a smaller correction for CBD. The preoperative |CBD| did not differ between these groups.

ASD research has focused on the clinical impact of disorders affecting sagittal spine– pelvic parameters and their radiological assessment. The SRS–Schwab classification system is based primarily on the influence of sagittal radiographic variables [17]. Obtaining optimal corrective alignment through ASD surgery prevents postoperative mechanical complications [24]. Glassman et al. reported that a global CM > 40 mm is associated with worse pain and function in unoperated patients with ASD [5]. A recent study has also shown that CM correction can eliminate painful symptoms, such as rib–iliac impingement pain and gait disturbance [25]. However, despite these reports, until now, studies have focused on sagittal correction rather than the correction of CM.

Preoperative coronal alignment can impact the incidence of postoperative coronal imbalance in patients with ASD. Bao et al. reported that patients with preoperative CBD > 30 mm and trunk shifting to the convex side of the main curve are at greater risk for postoperative coronal imbalance than those with trunk shifting to the concave side of the main curve [8]. Building on Bao’s classification, Obeid et al. proposed a similar but more comprehensive classification that used 20 mm of preoperative CBD as the threshold for imbalance and added various modifiers, such as flexibility of the main curve and/or lumbosacral fractional curve. They demonstrated that correction should be obtained at the apex of the main curve for concave CM (type 1) but at the lumbosacral junction for convex and convex-like CM (type 2). Obeid et al. also stated that overcorrection of the main lumbar curve should be avoided in coronally aligned spines (type 0) to prevent the postoperative onset of CM [9].

It has been suggested that patients with different preoperative CM types can have different surgical outcomes. Zhang et al. showed that the incidence of postoperative coronal imbalance was significantly higher in patients with preoperative Obeid type 2 than in those with preoperative Obeid type 1 [26]. They examined an ASD patient who underwent deformity correction surgery with a posterior-only approach in their case series. Several studies have reported the correction of ASD using LLIF, a frequent indication for LLIF [27,28,29]. Anterior release using LLIF techniques can correct misalignment in the coronal and sagittal planes and may even restore the disc height [30]. However, few studies have reported the details of postoperative outcomes for using LLIF to correct CM in patients with ASD.

Therefore, we focused on whether the LLIF procedure with posterior fusion would change the radiological outcomes of CM correction. About half of our patients with ASD were classified as having type L using the SRS–Schwab system, and CM was found in 29 of these patients (85.3%). Most patients showed the Obeid type 1A curve pattern. By contrast, patients with Obeid type 2A showed a smaller ΔCBD postoperatively, and many still exhibited CM after surgery.

All patients received LLIF as the spinal correction surgery in this study. In the sagittal plane, it was suggested that CM types did not affect SRS–Schwab assessment factors ΔSVA, ΔPT, and ΔPI-LL. LLIF and posterior corrective fusion corrected the preoperative CM in nearly 70% of patients with Obeid type 1A. However, using LLIF to correct the CM deformity in the coronal plane in a patient with Obeid type 2A may worsen the main lumbar curve by causing malalignment toward the convex side (Figure 5 and Figure 6).

L4 tilt was larger before and after surgery in patients with Obeid type 2 CM than in those with Obeid type 1 CM, and there was no change in CBD before and after surgery in patients with Obeid type 2 CM.

It has been noted that when performing corrective surgery on patients with Obeid type 2 CM, sufficient correction of the coronal L4 tilt and the main lumbar curvature is required to achieve coronal balance [22]. It was found that even if anterior release using LLIF was performed in the same way as the posterior-only approach, the results were similar in patients with Obeid type 2 CM. In patients with Obeid type 2 CM, three-dimensional correction using LLIF and simple posterior corrective fusion is complex, and the “kickstand rod” (KR) technique may be helpful. The KR technique is a novel surgical procedure developed for CM correction in patients affected by ASD with sagittal imbalance [31,32]. Future studies should examine the use of this procedure for ASD patients with Obeid type 2 CM.

The limitations of this study were as follows. First, it was retrospective with a small sample size, and other confounding factors, such as selection bias, including surgical technique, might have affected the results. Second, this study did not involve long-term follow-up or functional scores that reflect the health-related QOL. As the title suggests, our study was primarily a radiological evaluation. For this reason, it is difficult to evaluate how the clinical results of CM are reflected in patients with complications, such as PJF and RB, when conducting clinical evaluations. However, a past report has found that CM does not affect QOL [16]. Studies with a larger sample size, long-term follow-up, and clinical evaluation are needed. Third, the length of hospital stay may be longer. This may be due to the Japanese medical insurance system and culture and may not be simply compared to other countries. Finally, when interpreting correlations, it is difficult to identify possible causal relationships between scoliosis, CBD, and related radiographic parameters. It is impossible to conclude that coronal L4 tilt and CBD cause preoperative and postoperative CM. Despite these limitations, the results of the current study show a lower rate of improvement in CM after anteroposterior staged surgery with concomitant LLIF in patients with Obeid type 2 ASD than those with Obeid type 1 ASD.

## 5. Conclusions

We investigated the change in CM after anteroposterior staged surgery with LLIF in patients with ASD and compared patients grouped according to the preoperative lumbar curve. As reported previously by others, patients with Obeid type 2 CM had worse outcomes than patients with Obeid type 1 CM. Obeid type 2 CM may require different correction techniques, such as the KR technique, to improve the CBD.

## Figures and Tables

**Figure 1 jcm-12-02389-f001:**
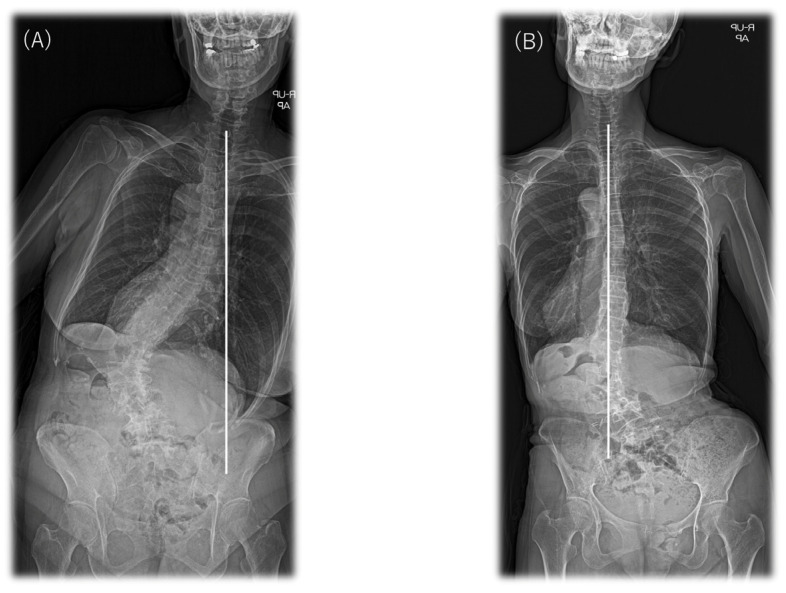
Two types of CM (type 1 and type 2) in patients with ASD classified according to the Obeid CM system. (**A**) An example of concave (type 1) and (**B**) convex (type 2) CM. CM, coronal malalignment.

**Figure 2 jcm-12-02389-f002:**
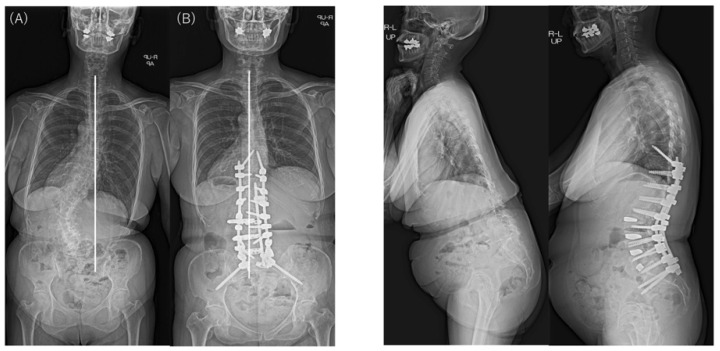
Preoperative and postoperative standing radiographs showing coronal and sagittal correction using anteroposterior staged surgery for ASD in a 63-year-old woman with type 1 CM. CBD changed from 41.6 mm preoperatively (**A**) to –3.2 mm postoperatively (**B**), with the C7 plumb line on the opposite side of the CBD. Sagittal alignment improved significantly, and postoperative CM also improved to CBD < 20 mm. CM, coronal malalignment; CBD, coronal balance distance.

**Figure 3 jcm-12-02389-f003:**
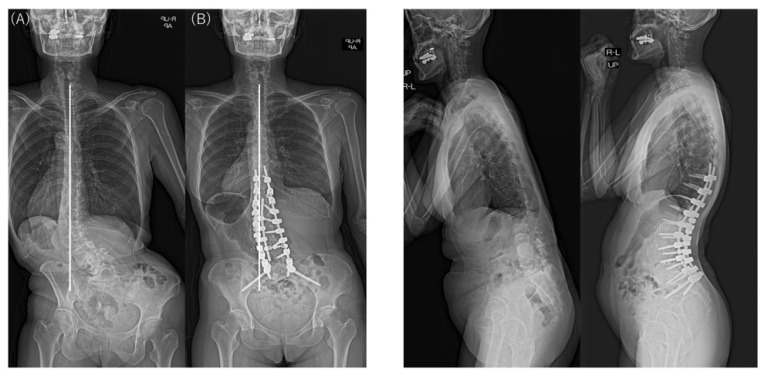
Preoperative and postoperative standing radiographs showing coronal and sagittal correction using anteroposterior staged surgery for ASD in a 55-year-old woman with type 2 CM. CBD changed from –86.7 mm preoperatively (**A**) to –38.8 mm postoperatively (**B**), with the C7 plumb line on the same side as the CBD. Sagittal alignment improved significantly, but CBD remained at ≥20 mm postoperatively. CM, coronal malalignment; CBD, coronal balance distance.

**Figure 4 jcm-12-02389-f004:**
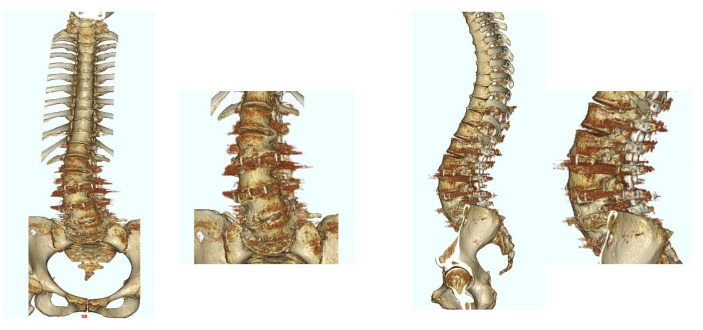
Postoperative 3D CT image of the same patient as in Figure 3. The patient underwent LLIF at L2/3, L3/4, and L4/5 using PEEK cages.

**Figure 5 jcm-12-02389-f005:**
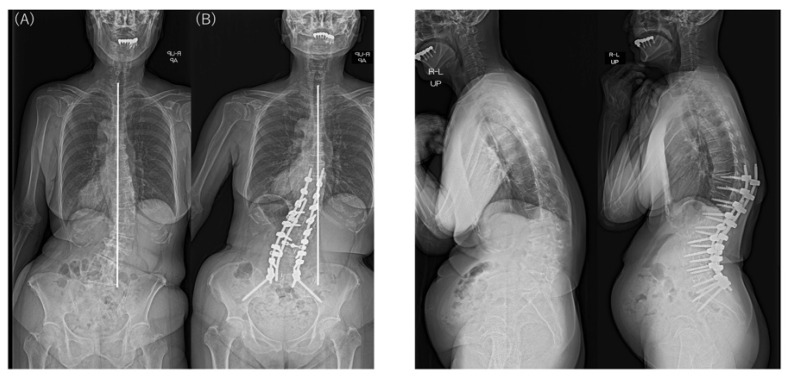
Preoperative and postoperative standing radiographs showing worsened CM after surgery in a 75-year-old woman with type 2 CM. CBD worsened from 38.1 mm preoperatively (**A**) to 67.7 mm postoperatively (**B**), with the C7 plumb line on the same side as the CBD. Sagittal alignment improved, but proximal junctional kyphosis was noted postoperatively.

**Figure 6 jcm-12-02389-f006:**
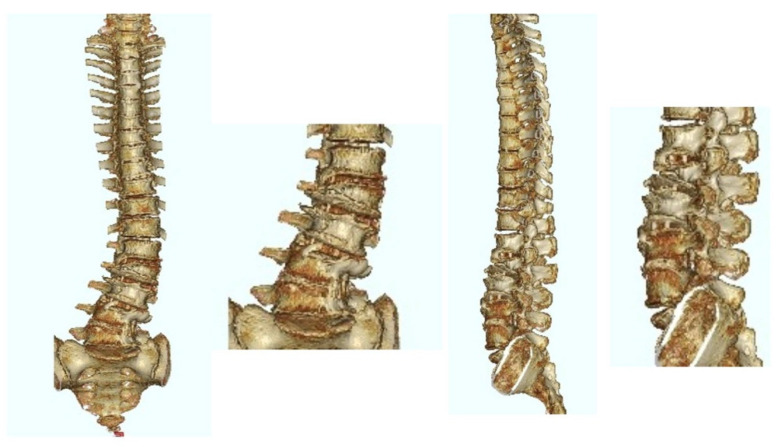
Postoperative 3D CT image of the same patient as in Figure 5. The patient underwent LLIF at L1/2, L2/3, and L3/4 using PEEK cages.

**Table 1 jcm-12-02389-t001:** The distribution of modifiers in the Type L according to Obeid-coronal. Malalignment subtype values are presented as number (%).

Subtype	Apex	No.
1A	L1 = 1 L2 = 9 L3 = 12 L4 = 1	23 (67.6)
1A1	L1 = 0 L2 = 3 L3 = 4 L4 = 0	7 (20.6)
1A2	L1 = 1 L2 = 6 L3 = 8 L4 = 1	16 (47.1)
1B	0	0 (0)
2A	L1 = 3 L2 = 1 L3 = 2 L4 = 0	6 (17.6)
2A1	L1 = 1 L2 = 0 L3 = 1 L4 = 0	2 (5.9)
2A2	L1 = 2 L2 = 1 L3 = 1 L4 = 0	4 (11.8)
2B	0	0 (0)
No coronal malalignment	L1 = 1 L2 = 4 L3 = 0 L4 = 0	5 (14.7)
Total	L1 = 5 L2 = 14 L3 = 14 L4 = 1	34 (100)

**Table 2 jcm-12-02389-t002:** Demographics of the two groups. Data presented as mean (SD) or number of patients (%). CBD, coronal balance distance, HT, height; BW, body weight; BMI, body mass index; LLIF, lateral lumbar interbody fusion; PSF, posterior spine fusion; EBL, estimated blood loss; SVA, sagittal vertical axis; PT, pelvic tilt; PI, pelvic incidence; LL, lumbar lordosis; UIV, upper instrumented vertebra; LIV, lower instrumented vertebra; HU, Hounsfield units. * Statistically significant.

Type L		Type 1A	Type 2A	ALL	*p*
No. of patients		23 (79.3)	6 (20.7)	29 (100)	
Age (yrs)		72.1(5.9)	70.8 (9.2)	71.8 (6.5)	0.683
60 y<		23 (100)	5 (83.3)	28 (96.6)	
Female, *n*		23 (100)	6 (100)	29 (100)	-
Follow-up periods (M)		25.3 (10.1)	29.2(9.3)	26.1 (9.9)	0.401
HT (cm)		150.1 (5.4)	151.1 (9.3)	150.3 (6.2)	0.742
BW (kg)		48.8 (9.4)	51.2 (11.3)	49.3 (9.7)	0.589
BMI (kg/m^2^)		21.6 (3.8)	22.3 (3.7)	21.7 (3.7)	0.663
Subcategories	PI-LL: 0	0 (0)	0 (0)	0 (0)	0.043 *
	PI-LL: +	4 (17.4)	0 (0)	4 (13.8)	
	PI-LL: ++	19 (82.6)	6 (100)	25 (86.2)	
	SVA: 0	0 (0)	0 (0)	0 (0)	0.472
	SVA: +	2 (8.7)	0 (0)	2 (6.9)	
	SVA: ++	21 (91.3)	6 (100)	27 (93.1)	
	PT: 0	1 (4.3)	1 (16.7)	2(6.9)	0.396
	PT: +	2 (8.7)	1 (16.7)	3 (10.3)	
	PT: ++	20 (87.0)	4 (66.7)	24 (82.8)	
CBD shift type	+ (right)	16	3	19	0.388
	− (left)	7	3	10	
Apex of main curve	L1	1	3	4	0.114
	L2	9	1	10	
	L3	12	2	14	
	L4	1	0	1	
Mean segments fused		9.4 (3.0)	9.3 (2.0)	9.4 (2.8)	0.939
Mean LLIF segments fused		3.0 (0.8)	2.7 (0.5)	3.0 (0.7)	0.268
UIV (T1-T5/T6-T8/T9-12/L1-L2)		6/1/14/2	1/1/4/0	7/2/18/2	0.961
LIV (fixed to sacrum/not fixed to sacrum)		22/1	6/0	28/1	0.618
Average HU values	UIV	142.7 (45.4)	128.1 (57.9)	139.6 (47.5)	0.517
	UIV +1	145.7 (41.0)	138.4 (51.9)	145.2 (42.7)	0.718
	UIV +2	149.4 (48.3)	140.9 (52.2)	147.6 (48.3)	0.709
Operation time (min)	1st (LLIF)	128.5 (46.3)	117.2 (44.6)	126.1 (45.4)	0.596
	2nd (PSF)	346.7 (85.7)	374.3 (63.7)	352.4 (81.4)	0.468
	Total	469.6 (110.6)	491.5 (103.7)	474.1 (107.8)	0.666
Intraoperative EBL (mL)	1st (LLIF)	63.9 (47.4)	81.2 (77.4)	67.6 (53.9)	0.496
	2nd (PSF)	666.2 (624.7)	855.7 (578.7)	705.4 (610.4)	0.508
	Total	727.3 (631.3)	936.8 (639.0)	770.6 (627.3)	0.476
Length of stay (days)		28.4 (6.4)	29.2 (3.5)	28.5 (5.8)	0.766
Proximal junctional failure (yes/no)		5/18	0/6	5/24 (17.2)	-
Rod breakage (yes/no)		8/15	0/6	8/21 (27.6)	-
Revision surgery (yes/no)		8/15	0/6	8/21 (27.6)	-

**Table 3 jcm-12-02389-t003:** Preoperative and postoperative spinopelvic parameters of the two groups. Data presented as mean (SD) or number of patients (%). CBD, coronal balance distance, SVA, sagittal vertical axis; PI, pelvic incidence; PT, pelvic tilt; SS, sacral slope; LL, lumbar lordosis; TK, thoracic kyphosis; PJA, proximal junctional angle. † Comparison with pre op, ‡ comparison between groups. * *p* < 0.05 statistically significant.

Type L		Type 1A	Type 2A	ALL	*p* ‡
No. of patients		23 (79.3)	6 (20.7)	29 (100)	
Major Cobb angle (°)	Preop	50.0 (11.4)	44.3 (11.4)	48.8 (11.5)	0.286
	Last	12.9 (11.9)	14.2 (8.0)	13.2 (11.1)	0.810
	*p* †	<0.001 *	0.002 *	<0.001 *	
│CBD│ (mm)	Preop	78.4 (51.6)	46.9 (21.7)	71.9 (48.5)	0.159
	Last	22.8 (28.0)	38.5 (20.4)	26.0 (27.0)	0.211
	*p* †	<0.001 *	0.506	<0.001 *	
Coronal L4 tilt (°)	Preop	16.0 (6.8)	28.7 (10.3)	18.6 (9.1)	0.001 *
	Last	7.3 (3.4)	14.0 (5.1)	8.7 (4.6)	0.001 *
	*p* †	0.002 *	0.008 *	<0.001 *	
Coronal L5 tilt (°)	Preop	11.7 (7.7)	17.4 (1.3)	12.9 (7.3)	0.002 *
	Last	6.7 (3.9)	9.5 (2.4)	7.3 (3.8)	0.106
	*p* †	<0.001 *	0.001 *	<0.001 *	
SVA (mm)	Preop	168.0 (75.9)	172.9 (35.1)	169.0 (69.0)	0.879
	Last	36.3 (56.3)	53.3 (39.9)	36.9 (53.4)	0.494
	*p* †	<0.001 *	0.001 *	<0.001 *	
PI (°)	Preop	50.2 (6.7)	51.3 (8.6)	50.4 (7.0)	0.739
	Last	51.9 (10.4)	50.8 (7.3)	51.0 (9.8)	0.823
	*p* †	0.520	0.794	0.593	
PT (°)	Preop	34.7 (7.0)	31.6 (8.6)	34.1 (7.3)	0.361
	Last	18.7 (11.1)	16.8 (8.6)	17.6 (10.2)	0.687
	*p* †	<0.001 *	0.006 *	<0.001 *	
SS (°)	Preop	15.5 (8.1)	19.7 (6.1)	16.3 (7.8)	0.245
	Last	33.1 (10.3)	34.1 (8.6)	33.4 (9.8)	0.835
	*p* †	<0.001 *	0.001 *	<0.001 *	
LL (°)	Preop	4.6 (19.9)	3.5 (4.0)	4.4 (17.7)	0.809
	Last	55.0 (12.2)	61.5 (7.2)	56.8 (11.4)	0.224
	*p* †	<0.001 *	<0.001 *	<0.001 *	
TK (°)	Preop	25.9 (17.4)	7.5 (11.5)	22.1 (17.9)	0.021 *
	Last	44.3 (13.2)	40.0 (3.1)	42.9 (11.5)	0.156
	*p* †	<0.001 *	0.002 *	<0.001 *	
PI-LL (°)	Preop	45.6 (22.1)	47.8 (10.7)	46.1 (20.1)	0.817
	Last	−3.1 (14.3)	−10.7 (12.7)	−5.7 (13.7)	0.249
	*p* †	<0.001 *	<0.001 *	<0.001 *	
PJA (°)	Preop	5.2 (5.3)	1.4 (5.4)	4.5 (5.4)	0.125
	Last	17.9 (10.8)	19.0 (7.6)	17.7 (10.3)	0.825
	*p* †	<0.001 *	0.007 *	<0.001 *	

**Table 4 jcm-12-02389-t004:** Parameter changes of patients in the two groups at last follow-up. Data presented as mean (SD) or number of patients (%). CBD, coronal balance distance, SVA, sagittal vertical axis; TK, thoracic kyphosis; LL, lumbar lordosis; PI, pelvic incidence; PT, pelvic tilt; SS, sacral slope; PJA, proximal junctional angle. † Comparison between groups. * *p* < 0.05 statistically significant.

Type L	Type 1A	Type 2A	ALL	*p* †
No. of patients	23 (79.3)	6 (20.7)	29 (100)	
Δ Major Cobb angle (°)	−37.1 (14.2)	−30.1 (12.5)	−35.6 (13.9)	0.284
Δ│CBD│(mm)	76.6 (50.7)	24.1 (15.0)	65.8 (50.3)	<0.001 *
ΔCoronal L4 tilt (°)	−8.6 (5.1)	−14.7 (8.4)	−9.9 (6.3)	0.142
ΔCoronal L5 tilt (°)	−5.0 (6.9)	−7.9 (3.0)	−5.6 (6.4)	0.326
ΔSVA (mm)	−135.3 (80.4)	−119.6 (42.5)	−132.1 (73.7)	0.650
ΔPI (°)	0.9 (6.6)	−0.5 (4.1)	0.6 (6.1)	0.635
ΔPT (°)	−16.9 (9.7)	−14.9 (7.9)	−16.5 (9.2)	0.633
ΔSS (°)	17.8 (9.9)	14.4 (4.5)	17.1 (9.1)	0.418
ΔLL (°)	50.9 (22.1)	58.0 (4.1)	52.4 (19.9)	0.162
ΔTK (°)	17.7 (15.1)	32.5 (12.7)	20.8 (15.7)	0.037 *
ΔPI-LL (°)	−50.0 (22.6)	−58.5 (7.5)	−51.8 (20.6)	0.146
ΔPJA (°)	12.1 (10.1)	17.6 (9.9)	13.2 (10.1)	0.245

**Table 5 jcm-12-02389-t005:** Number of patients with improved CM after surgery. CBD, coronal balance distance.

		Postoperative		
No. of Patients		CBD < 20 mm	CBD ≥ 20 mm	Improvement Rate
Type 1A	23	16	7	69.6%
Type 2A	6	1	5	16.7%

## Data Availability

The data presented in this study are available on request from the corresponding author. The data are not publicly available due to privacy or ethical restrictions.

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
