# Peer review of "Postoperative Radiological Improvement after Staged Surgery Using Lateral Lumbar Interbody Fusion for Preoperative Coronal Malalignment in Patients with Adult Spinal Deformity"

_jcm, 2023, doi:10.3390/jcm12062389_

Round 1
Reviewer 1 Report
congratulations to the authors for an outstanding paper.
i read it will considerable interest, as i have a special passion for the coronal plane in adult spinal deformity.
the authors present a novel evaluation of the utility of a specific surgical strategy (multi-level direct lateral interbody fusion, multi-level thoracolumbar PSF, and L5-S1 TLIF) for coronal realignment in ASD. This is a very important topic and one that clinically is very relevant to practicing physicians.
The introduction, methods, results, and discussion are very well written and concise.
My only recommendation is to add to Table 1 the type of curve (flexible, apex of cobb) next to each obeid type (i.e. type 1A1 = X; type 1A2 = Y, 1B = Z; etc.). this will make readability easier.
Author Response
Reviewer #1: congratulations to the authors for an outstanding paper.
I read it will considerable interest, as i have a special passion for the coronal plane in adult spinal deformity.
the authors present a novel evaluation of the utility of a specific surgical strategy (multi-level direct lateral interbody fusion, multi-level thoracolumbar PSF, and L5-S1 TLIF) for coronal realignment in ASD. This is a very important topic and one that clinically is very relevant to practicing physicians.
The introduction, methods, results, and discussion are very well written and concise.
My only recommendation is to add to Table 1 the type of curve (flexible, apex of cobb) next to each obeid type (i.e. type 1A1 = X; type 1A2 = Y, 1B = Z; etc.). this will make readability easier.
Response> We sincerely thank the reviewer for this kind, insightful comment. Following the referee's instructions, we added curve Apex levels to Table 1.
Reviewer 2 Report
Dear, authors
My suggest and comment are below.
1. In Figure 1: Author should be demonstrated by using C7 plumb line for represent Obeid CM classification instead of using CSVL.
2. In surgical procedure and Table 2: Author should give more detail in type of bone substitutes (BMP-2 or HA) in LLIF surgery between these two groups because it has effect on rod breakage and revision surgery. Also, Author should give more detail on grating which use during TLIF on L5-S1. Why did author use TLIF for L5-S1 instead of ALIF L5-S1 with more powerful in correct sagittal imbalance?
Author Response
My suggest and comment are below.
- In Figure 1: Author should be demonstrated by using C7 plumb line for represent Obeid CM classification instead of using CSVL.
Response> We would like to thank you for your careful review of our work. We have revised the figure according to your comments.
- In surgical procedure and Table 2: Author should give more detail in type of bone substitutes (BMP-2 or HA) in LLIF surgery between these two groups because it has effect on rod breakage and revision surgery.
Response> We sincerely thank the reviewer for this kind, insightful comment. We added the following comments about the LLIF cage in surgical procedures. “demineralized bone matrix (Grafton,
Also, Author should give more detail on grating which use during TLIF on L5-S1. Why did author use TLIF for L5-S1 instead of ALIF L5-S1 with more powerful in correct sagittal imbalance?
Response>We sincerely thank the reviewer for this kind, insightful comment. We agree with the reviewer's opinion. However, LLIF could not be performed at L5–S1 in general; thus, TLIF was usually performed on both sides using the posterior approach. Recently, OLIF51 has been introduced in Japan, but a license is required in Japan. Against this background, L5/S1 is corrected with TLIF at our facility. Thank you.
We have added the following limitations to the problems with these procedures. “The limitations of this study were as follows. First, it was retrospective with a small sample size, and other confounding factors, such as selection bias, including surgical technique, might have affected the results.”